# EDEM Simulation Study on the Performance of a Mechanized Ditching Device for Codonopsis Planting



**Dejiang Liu** [1,2], **Yan Gong** [1,*], **Xuejun Zhang** [2], **Qingxu Yu** [1], **Xiao Zhang** [1], **Xiao Chen** [1] **and Yemeng Wang** [1]

[1] Nanjing Institute of Agricultural Mechanization, Ministry of Agriculture and Rural Affairs, Nanjing 210014, China

[2] College of Mechanical and Electrical Engineering, Xinjiang Agricultural University, Urumqi 830052, China

[*] Correspondence: gongyan@caas.cn

**Abstract:** Codonopsis pilosula is cultivated mainly in sandy soils, especially in the Dingxi area of the Gansu province, northwest China. They are mainly planted in hilly areas, where large machines cannot reach easily. Codonopsis pilosula transplanting has been adopted with a conventional flat planting way, film mulching and seedling outcrops. While its planting requires opening a shallow ditch, short operation cycle, considerable labor intensity and is in large demand, a simulation analysis was performed according to specific tillage resistance, helpful in the optimization design in later stages and the improvement of a domestic ditching plow's performance. This paper studies the simulation performance of an adjustable trench plough and analyzes the orthogonal test and Design-Expert response surface of the data. EDEM simulation software was used to analyze the traction resistance of the furrow ditching mechanism in the ditching process. The results show that the traction resistance increased from 1751.31 N to 2197.31 N as the simulation working speed increased from 0.9 m/s to 1.5 m/s, 1.25 times higher than the former. It indicates that speed had significant effects on traction resistance. With the increase in working speed, the furrow traction resistance and specific consumed power were increased considerably. Using the stability coefficient of the ditching depth and consistency coefficient of ditching bottom width as test indices, speed and angle of furrowing as impact factors, Box–Behnken orthogonal experiment design method, and establishing test indices and test factors, the regression model between the analysis of the influence of the test index, comprehensive agronomic requirements and MATLAB factor optimization of the experiment, the optimized operation parameter combination was obtained: the forward speed was 0.9 m/s, the angle of ditching was 35° and the stability coefficient of the ditching depth and consistency coefficient of ditch bottom width were 97.57% and 98.03%, respectively. The data after the simulation comparison test can provide a design reference for the domestic small trench operation.

**Keywords:** Codonopsis planting; ditching device; working parameters; EDEM simulation; orthogonal test; data analysis and discussion

## 1. Introduction

Traditional Chinese medicine is one of the characteristic industries in the Gansu province, China. Since national industrial policy transferred to characteristic industries, the areas of traditional Chinese medicine are gradually increasing year by year in the Gansu province. The planting for Chinese medicine is an effective way for vast rural areas and farmers to avoid poverty and increase their wealth. The traditional Chinese medicine cultivation area has a large scale and high visibility. In the Gansu province, Radix Codonopsis accounts for 60% of the whole country. Codonopsis pilosula is a traditional Chinese medicine, also known as "tiaodang", belonging to herbs of the Platycodon family. It has a soft and solid texture, slightly fissured section and yellow and white skin. It enhances the body's stress ability functions, immune functions, delaying senescence, and has anti-ulcer, anti-tumor, anti-bacterial, etc., properties. The shape of Codonopsis pilosula

is oval, and its long and short axis is about 1.2 mm and 0.68 mm, respectively. Codonopsis pilosula planting can be achieved through sowing or seedling transplanting. The sowing process has a long growing cycle (3–5 years), for which is difficult to achieve scale benefits. The seedling transplanting method is usually adopted with the film mulching model and seedling outcrops. In this method, Codonopsis pilosula seedlings are inside the soil, but outside the film. It ensures a smooth emergence and prevents the occurrence of burning seedlings. It prevents drought and waterlogging, and also provides convenience for mechanical harvest, enhancing the commodity rate, and achieving production goal and thus increased income. For Codonopsis-membrane-covered outcrop cultivation, a ditch is needed first (the ditch type should be an inclined ditch), and then codonopsis is planted obliquely, then covered with the soil film, and finally leveled, which is the entire process to complete the planting of Codonopsis. Based on the cultivation agronomy of codonopsis pilosula with the film-covered outcrop, there is no mechanical operation in the ditching cultivation process to match it, and the traditional manual operation is adopted, which has high labor intensity and low production efficiency, seriously restricting the development of the traditional Chinese medicinal materials industry in Gansu [1–3].

Northwest China has a vast area, dry climate and large spatial difference of soil. When the ditching device is in direct contact with the soil, it is subject to the random resistance and friction of the soil and the vibration generated by the machine, which can easily cause the deformation of the ditching plow structure and affect the ditching accuracy. At present, domestic and international scholars have carried out relevant research on the force analysis of ditching devices in soil, analyzing soil movement characteristics, the influence law of ditching power consumption, etc. [4–6]. Wang Shaowei et al. designed the ditching components of the Mountain Orchard ditching machine, optimized the structure parameters of the ditching blade by simulation tests [7] and analyzed the influence of operation parameters on ditching power consumption. Zhang Yongliang et al. applied the theory of throwing soil to analyze the process of throwing soil, finding the trajectory equation of the thrown soil particles, and establishing a mechanics model of soil discrete element contact studying its throwing performance with the method of experimental verification [8]. Kang Jianming et al. established a finite element model for a soil ditching cutter head by using the smoothed particle hydrodynamics method and obtained the variation law of power consumption of the slotting cutter head in the soil cutting process through simulation analysis [9]. Barr et al. studied the effects of different openers on soil disturbance and soil pressure at different speeds through the orthogonal test and reduced soil disturbance by increasing the ditching operation speed [10].

In view of the above problems, based on plough-type opener trenching as the research object, through EDEM simulation and orthogonal experiment analysis, the analysis of different furrow forms of change and the influence of structure parameters on traction resistance are investigated, seeking to arrive at the soil components combined furrowing device for low consumption, as well as the theory of drag reduction design in order to obtain a better drag reduction characteristic form of furrowing. It provides a design reference for domestic small ditching operation.

## 2. Materials and Methods

### 2.1. The Dimensions of the Entire Structure and the Working Principle of Double-Side-Flip Plough Ditching Device

2.1.1. Dimensions of the Entire Structure

Based on the planting agronomic requirements of codonopsis pilotica's oblique ditching, a double-sided inverted ditching plough that can open oblique ditches was designed. The size diagram is shown in Figure 1. The length of the plough wall is 35 cm, the ground clearance is 37 cm, the section of the plough column is a square with a width of 6 cm, the total length of the plough blade and the plough bracket is 34 cm, the width of the plough bracket is 18 cm, and the length from plough tip to the end of the plough bracket is 36 cm. The adjustable double-sided inverted ditching plough can complete ditching, soil covering,

ploughing and other operations. The ploughshare can be turned around. The direction and angle of soil turning can be adjusted [11,12].

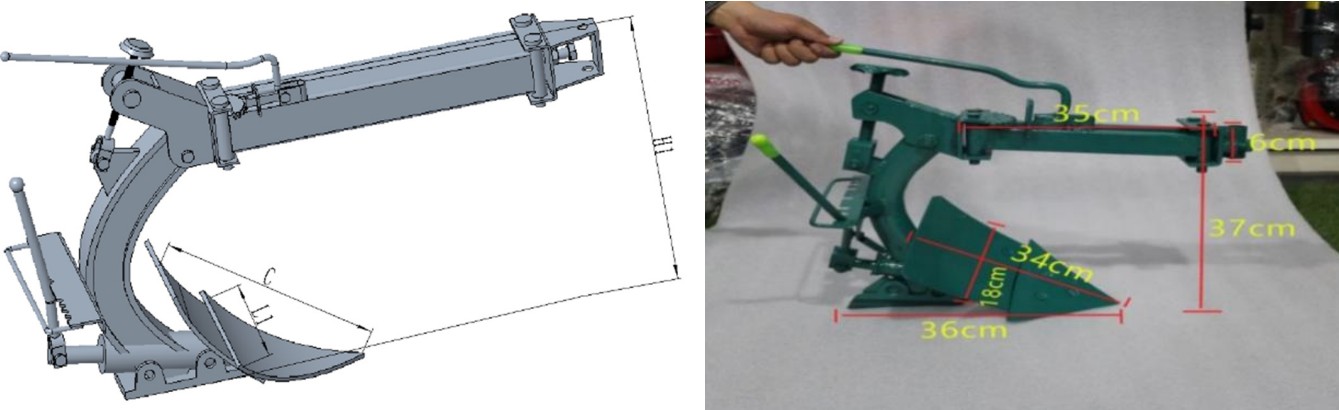

**Figure 1.** Overall dimension drawing of the double-sided turning plough ditching device.

2.1.2. Working Principle of Double-Side-Flip Plough Ditching Device

The power part equipped with the micro-tiller during the operation is shown in Figure 6. The double-side-flip plough ditching device is mainly composed of the main frame, plough blade, plough wall, tillage depth's adjusting rod and angle's adjusting rod, etc. The plough blade is installed at the bottom of plough. It cuts the soil laterally and keeps the same level at the bottom of trench. To dig an oblique flat trench horizontally, An adjustable rod is used to adjust the soil-insert depth. Once the planting ditch is finished, the seedlings are covered, and then another furrow is ditched. Using another furrow's soil to cover the prior furrow's seedlings, the process continues until we finish all ditching and soil covering The ploughshare can be turned around towards the left or right direction. The structure is simple, compact and flexible, and is suitable for small plot operation.

*2.2. Planting Agronomic Requirements*

During the survey, we found that, in actual production, Codonopsis pilosula are transplanted obliquely, as is shown in diagram 1. For this, an oblique furrow was dug, achieving a bottom depth at 10–15 cm. The seedlings were placed on one of the sloping surfaces evenly (angle of sloping surface is 10–40°). The distance between each seedling was 5–8 cm, and the top of seedling was 3–4 cm above the ground. We filled the soil after the seedlings were placed, leveling afterwards. The row spacing was 30 cm.

As shown in Figure 2b, the overall furrow was dug firstly, and the average depth was 8–10 cm. Then, the seedlings were put horizontally, as shown in Figure 2c. The distance between seedlings was 10–12 cm. The mulching film was covered along the red line. The edge of plastic film was 2–3 cm from the tip of seedling, which shows the tip outside the film. In the end, we covered the soil, as shown in Figure 2d. The depth of soil covering was 3–5 cm. The width and thickness were 900 mm and 0.012 mm, respectively, for each ridge.

*2.3. Design Analysis of Key Mechanisms*

2.3.1. Design of Double-Flip Plough Ditching Device with a Tilting-Plough Blade

In the tilting plough of the double-flip plough ditching device, the soil was spilled on the side of the plough body, which was convenient for ground grading lately and offset the lateral force of the plough body. Firstly, the curved wire was drawn, as shown in Figure 3a. The curved wire refers to the curve where the plough surface intersects with the vertical plane. In order to improve the soil flip effect, the plane of the curved wire is at the end of share blade, as shown in Figure 2b. The plough body surface was controlled by a straight element line, and the angle between the straight element line and the trench wall is the angle between the straight element line, which changes along the height of the plough body.

In the ploughshare part, it changes in a straight line and, in the plough wall part, it changes in a broken line, as shown in Figure 3c.

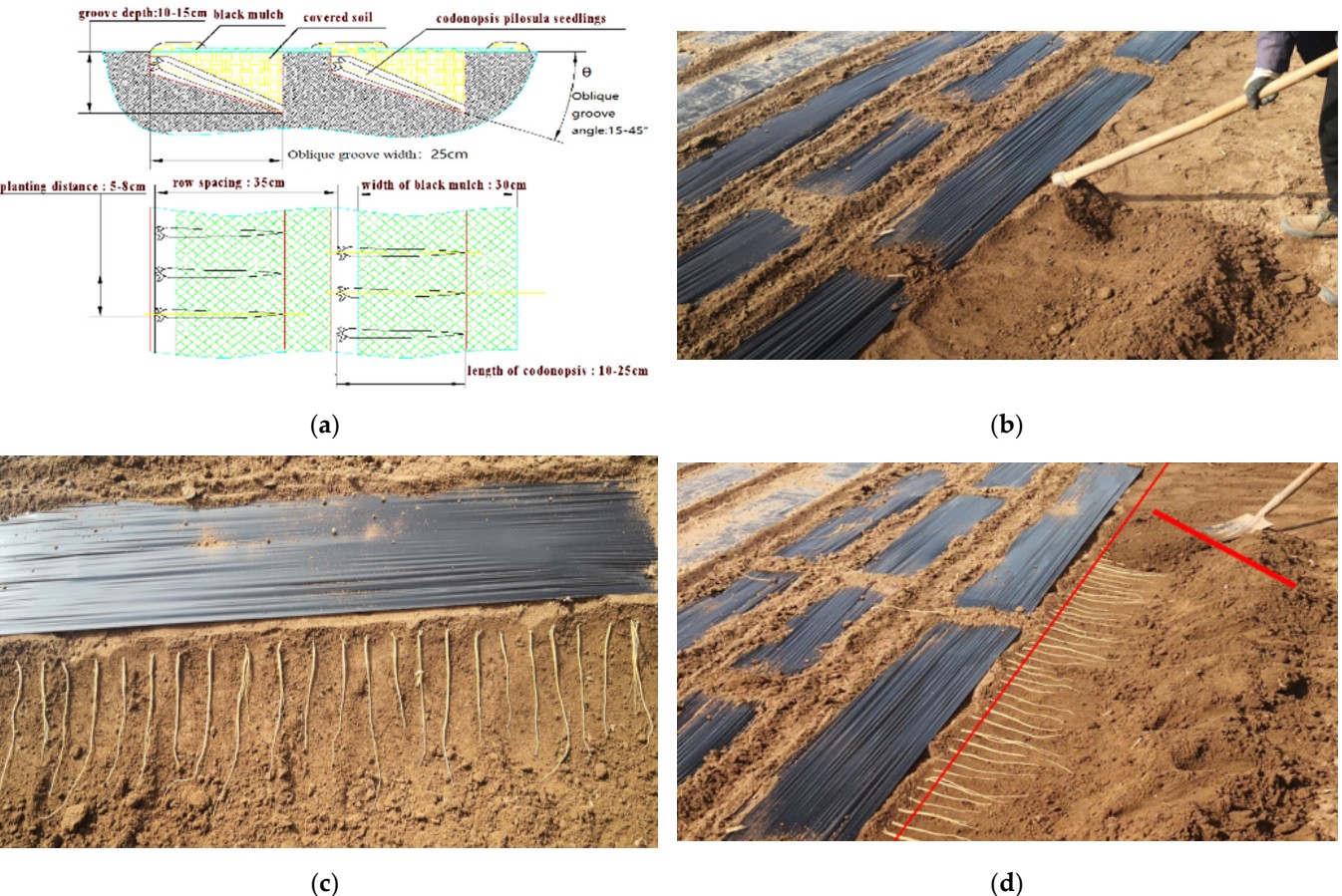

(**a**)

(**b**)

(**c**)

(**d**)

**Figure 2.** Codonopsis planting agronomic. (**a**) Codonopsis pilosula's planting agronomy. (**b**) Ditching. (**c**) Codonopsis transplanting seedlings. (**d**) Soil and film covering.

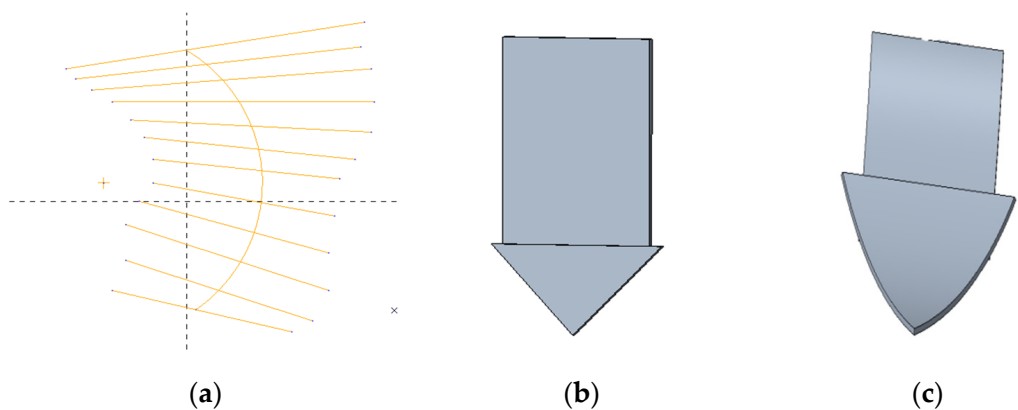

(**a**)                                          (**b**)                                          (**c**)

**Figure 3.** Codonopsis pilosula planting agronomy. (**a**) The plough plots the curve. (**b**) Straight plough. (**c**) The bent plough blade.

### 2.3.2. Plough Body Parameter Design

The adjustable ditching plough belongs to plough-type ditching device, with a simple structure, reliable work, strong ability of breaking soil into the soil, the heavy soil, wasteland and weeds of the plot have a strong adaptability, the ploughing angle and pushing angle of the plow body according to the structural parameters of the share plow design, the main design of the width of the plough body and the height of the plough body.

In the actual working process, the installation position of the ditching plough is directly in front of the intermediate gearbox, so the space is limited. The width of the plough body C should be larger than the width of the bottom of the intermediate gearbox box (300 mm). At the same time, considering that the plough body will tear the soil, the actual width of the plough is slightly larger than the width of the working groove surface $L_1$. That is, $L_1 < C \leq 350$ mm, the width C of the plough body should meet $L_1$.

The height of soil sliding along the plough is approximately equal to the depth of the working ditching $h$. In order to avoid soil sliding over the top of the plough, the soil should still be inside the plough after reaching the surface, and the height $H_1$ designed for the plough should meet the following requirements [9,10]:

$$H_1 \geq h = \frac{L_1 - A}{2} tan\theta$$

In the formula, $\theta$ is the ridge body accumulation Angle (°) after ridge initiation, which was annotated in Figure 2a; $H_1$ is the design height of plough body (mm); $L_1$ is the width of the working groove surface (mm); h is the working ditching depth (mm); $A$ is the width of the trench bottom (mm), which is 150 mm. Since the height of the plough from the ground can also be adjusted through the height adjustment holes on the screw and the support rod, after comprehensive consideration, the height $H_1$ of the plough was determined to be 370 mm. See Figure 1 for letter labeling in the formula. The main technical parameters of the machines and tools are shown in Table 1.

**Table 1.** The main technical parameters of the machines.

| Project | Technical Parameters |
|---|---|
| Hang way | Tractor traction type |
| Supporting power (Metric horsepower) | $\geq$30 |
| Machine weight (kg) | 60 |
| Outline dimensions (mm) Length × width × height | 450 × 300 × 390 |
| Ground speed (m/s) | 0.9–1.5 |
| Trenching depth (mm) | 80–100 |
| The stability coefficient of ditching depth (%) | $\geq$90% |
| Consistency coefficient of ditching bottom width (%) | $\geq$85% |
| Efficiency of operations (ha/h) | $\geq$0.5 |
| Leakage rate (%) | $\leq$4 |

*2.4. Discrete Element Simulation Analysis*

The EDEM discrete element soil simulation model was established and verified by the soil through test results. EDEM simulation was carried out on the plough surface parameters and operating speeds, and the change rule of the plough traction resistance under different operating speeds and the change in the plough parameter element line angle were obtained.

2.4.1. Establishment of Mechanical Models

Discrete element contact mechanical models can be divided into soft sphere model, as shown in Figure 4a, and hard sphere model, as shown in Figure 4b. Due to the relatively low movement speed of soil particles and plough body and the fixed adhesion between soil particles, HertzMindlin soft ball model was adopted in this paper [13,14]. The model mainly includes physical property parameters (tangential elastic coefficient $k_T$, normal elastic coefficient $k_N$, tangential damping coefficient $C_t$ and normal damping coefficient $C_n$) and geometric parameters, normal overlap $\delta_n$ and tangential overlap $\delta_T$.

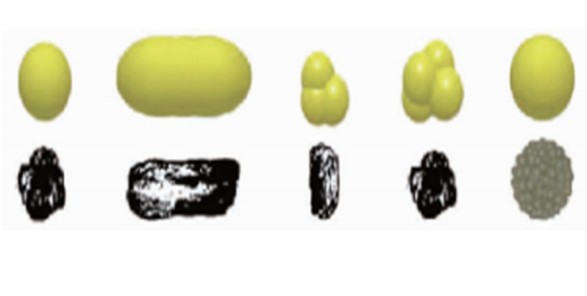 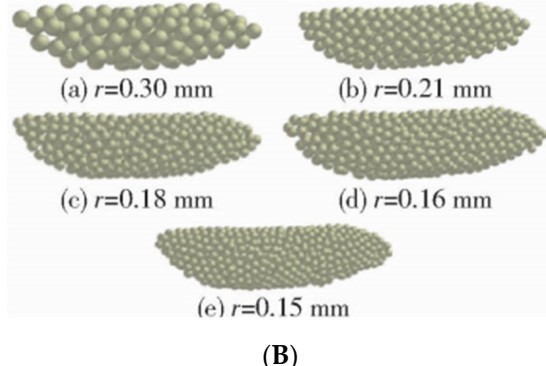

(**A**) (**B**)

**Figure 4.** Discrete element contact mechanical models. (**A**) Soft sphere model. (**B**) Hard sphere model.

### 2.4.2. Parameter Calibration

In the simulation model, tangential elastic coefficient $k_T$ and normal elastic coefficient $k_N$ were calculated according to elastic modulus E and shear modulus G [9]. Tangential damping coefficient $C_t$ and normal damping coefficient $C_n$ were calculated according to the corresponding normal and tangential elastic coefficients, material density $\rho$ and recovery coefficient e [13]. Tangential contact force $F_t$ was calculated according to the Mindlin theory [14], and normal contact force $F_n$ was calculated according to the Hertz theory. According to the Hertz contact theory [14–16], particles are regarded as isotropic materials in the process of particle collision, and the relationship between shear modulus $G$ and elastic modulus $E$ is:

$$G_j = \frac{E_j}{2(1+v_j)}$$
$$(j = 1, 2)$$

where,

$G_j$—shear modulus of particle, Pa;
$V_j$—Poisson's ratio of particle, %;
$E_j$—elastic modulus of particle, Pa.

### 2.4.3. Calibration Model

In Figure 5a, which shows that the soil is composed of solid particles, the joint strength between the soil particles is far less than the strength of the soil itself, so under the action of external forces between the soil particles, mutual dislocation occurs, causing one part of the soil relative to another part of the slide. The resistance of the soil to this slide is called the shear resistance of the soil. In Figure 5b, which shows when the soil accumulation, the slope of the pile (i.e., the angle between the pile and the ground) is called accumulation angle.

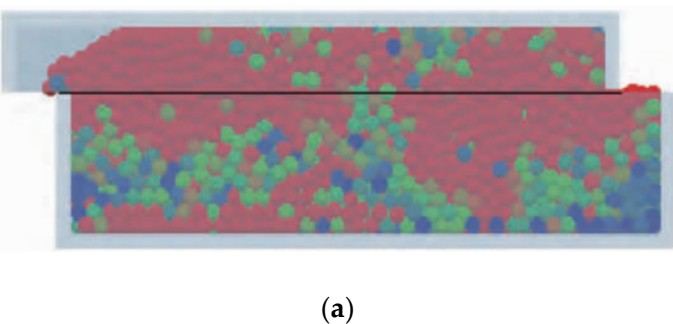 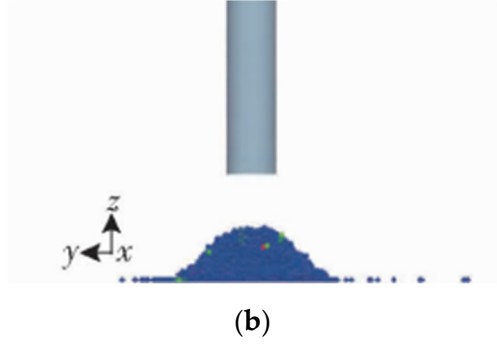

(**a**) (**b**)

**Figure 5.** Calibration model. (**a**) Soil self-shear model (time 1.5 s). (**b**) Soil particle calibration accumulation angle (time 3.5 s).

According to the relationship between shear modulus G, elastic modulus E and Poisson's ratio v, the third parameter can be determined by providing two of them. The soil particle radius is 5.5 mm, the static friction coefficient is 0.45, and the rolling friction coefficient is 0.21 [17]. In this paper, EDEM software was used to conduct simulation test research. The calibration model of soil parameters is shown in Figure 5: Figure 5a is the discrete element shear model of soil, Figure 5b is the soil accumulation angle model and the simulation parameters are shown in Table 2.

**Table 2.** The calibration model of the soil parameters.

| Parameter Attribute | Parameter | Value of Number |
|---|---|---|
| Soil particle properties | Density $\rho/(kg/m^3)$ | 2500 |
| | Shear modulus G/Pa | $2.2 \times 10^7$ |
| | Poisson's ratio v | 0.45 |
| | Soil particle radius r/mm | 5.5 |
| Material properties of the plough body | Density $\rho'/(kg/m^3)$ | $7.8 \times 10^3$ |
| | Shear modulus G'/Pa | $7.0 \times 10^{10}$ |
| | Poisson's ratio v' | 0.35 |
| Interaction between plough and soil | Quiet friction factor of soil particles and particles $\mu_1$ | 0.45 |
| | Dynamic friction factor between soil particles and particles $\mu_2$ | 0.21 |
| | Soil particle and particle collision recovery factor $R_1$ | 0.11 |
| | Quiet friction factor between the soil particles and the plough $\mu_3$ | 0.3 |
| | Dynamic friction factor between soil particles and plough $\mu_4$ | 0.26 |
| | Impact recovery factor between the soil particles and the plough $R_2$ | 0.2 |
| Other parameters | Acceleration of gravity $g/m \cdot s^{-2}$ | 9.8 |
| | Deep ploughing of the plough h/mm | 80–100 |
| | Width of the plough L/mm | 260 |
| | Number of soil particles N/number | 450,000 |

2.4.4. Establishment of Discrete Element Model for the Trenching Operation of the Trenching Plough

In order to improve the simulation accuracy of the discrete element, the adjustable ditching plow was imported into the EDEM simulation software. Based on the theory of soil stratification analysis, according to the overall size and operation parameters of the ditching device, a soil trough with a length of 4000 mm, a width of 900 mm and a soil layer thickness of 600 mm was established in EDEM software according to the simulation parameters in the table above, as shown in Figure 6. The thickness and section shape of the soil trough are consistent with the field situation. The soil thickness of plough layer is 200 mm, the soil thickness of plough bottom is 180 mm, and the soil thickness of bottom layer is 220 mm. The radius of soil particle unit in each layer is 5.5 mm.

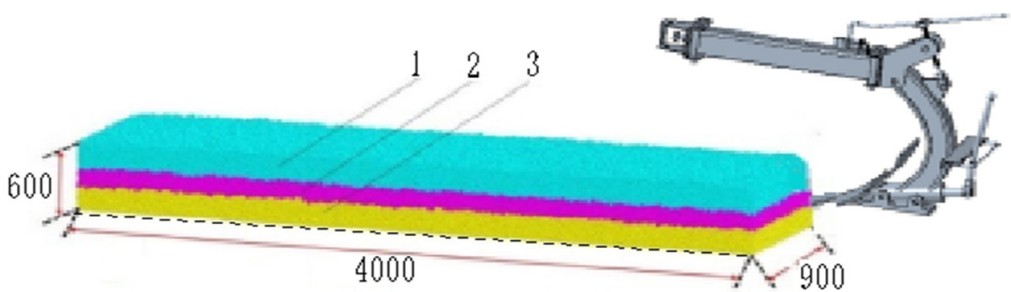

**Figure 6.** Simulation of the soil groove and ditching device geometric model. 1. Plough layer soil; 2. Plow sole; 3. Subsoil.

The following parameters are set in the Creator module of EDEM software. The forward speed of ditching was 0.9 m/s and 1.5 m/s, respectively, and the direction was

along the forward direction of ditching device. The angle between the ditching plough and the ground plane was set at 25° and 45°. The following parameters were set in the EDEM Simulator module: the time step was $2.1 \times 10^{-4}$ s, the action time was 4.5 s, the data storage interval was 0.06 s, and the grid cell size was 2.5 times the minimum radius of the soil particles. After all parameters were set, the ditching simulation was carried out in EDEM, and the operation status of the ditching device is shown in Figure 7.

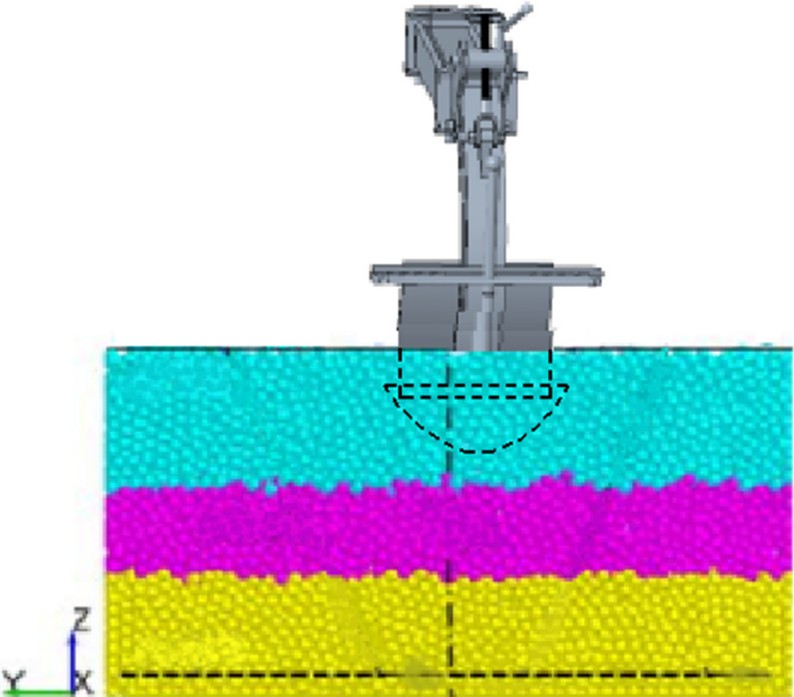

**Figure 7.** Ditching operation simulation diagram.

### 3. Results

*3.1. Single Factor Analysis of Simulation Data*

In order to study the influence of the change in the operating parameters of the adjustable tilting plough on the traction resistance and the evaluation index of ditching operation, the forward speed V and the angle θ between the ditching plough and the ground plane were taken as experimental factors. In the single factor analysis, the angle between the ditching plough and the ground plane was 25°, and the forward speed (0.9 m/s and 1.5 m/s) was changed. The influence of velocity change on traction resistance change was analyzed, as shown in Figure 8a,b.

According to the results of simulation, the plough body stress was introduced into the soil bin data, the data to map Figure 8a,b in 15 cm deep tillage, furrow plough Angle 25°, under different working speeds of traction resistance with the displacement variation, graph data for furrow plough and furrowing opener to open the soil to complete through the data between the data collection phases. As can be seen from the figure, the traction resistance of the trenching plough increases rapidly when it enters the soil, and then fluctuates back and forth within a certain range. The relationship between traction resistance and displacement is similar at different forward speeds. As the simulation speed increases from 0.9 m/s to 1.5 m/s, the traction resistance increases from 1751.31 N to 2197.31 N, which is 1.98 times that of the former, and the increase trend gradually increases. It shows that the speed has a significant influence on the traction resistance. With the increase in working speed, the traction resistance of the plough body and the corresponding power consumption increase significantly. In order to reduce power dissipation in actual production and cultivation, it is necessary to advance at a low speed. According to the analysis data in this paper, the best forward matching speed will be further studied in orthogonal experiments.

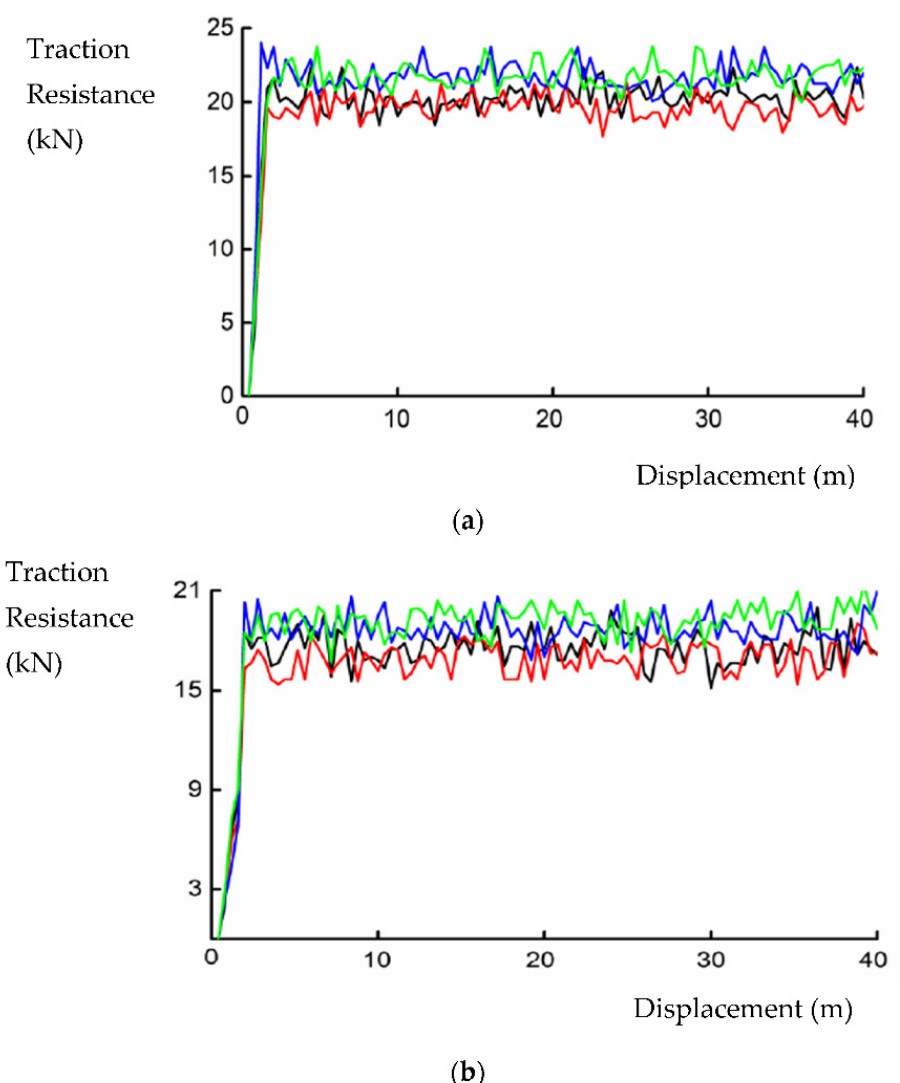

**Figure 8.** Traction resistance–displacement diagram of deep loose shovel at 15 cm tillage depth (the angle θ = 25° between the ditching plough and the ground plane). (**a**) Forward speed (0.9 m/s). (**b**) Forward speed (1.5 m/s).

*3.2. Orthogonal Test*

3.2.1. Test Design

In order to explore the interaction and influence law of the forward speed V and the angle θ between the ditching plough and ground plane on the stability coefficient of ditching depth and consistency coefficient of ditching bottom width, the Box–Behnken experimental design method was used to carry out the simulation test of the adjustable ditching operation parameters of Codonopsis pilotica, so as to study the optimal operation parameter combination of the ditching device. According to the test design, the forward speed and the angle between the ditching plough and the ground were set as independent variables X1 and X2, and the stability coefficient of ditching depth and the consistency coefficient of ditching bottom width were set as evaluation indexes Y1 and Y2. After the trench device is operated under certain working parameters, the degree to which the trench depth remains stable is expressed by the stability coefficient of ditching depth $Y_1$. After the trench device is operated under certain working parameters, the degree of consistency of the trench bottom width is expressed by the consistency coefficient of the trench bottom width $Y_2$. Additionally, the test factors and coding level of 2 factors 2 were specified, as shown in Table 3. The test results are shown in Table 4.

**Table 3.** Response surface test factor and levels.

| Level | Forward Speed $X_1$/(m·s$^{-1}$) | Angle between the Ditching Plough and the Ground $X_2$(°) |
|---|---|---|
| −1 low level | 0.9 | 15 |
| 0 middle level | 1.2 | 30 |
| 1 high level | 1.5 | 45 |

**Table 4.** Quadratic regression experimental design and response value.

| Test Number | Forward Speed | Angle | Stability Coefficient of Ditching Depth $y_1$/(%) | Consistency Coefficient of Ditching bottom Width $y_2$/(%) |
|---|---|---|---|---|
| 1 | 0 | 1 | 94.02 | 95.08 |
| 2 | 0 | −1 | 94.36 | 93.19 |
| 3 | 1 | 0 | 96.52 | 96.26 |
| 4 | 1 | −1 | 94.96 | 94.87 |
| 5 | 0 | 0 | 95.65 | 96.22 |
| 6 | −1 | 1 | 97.02 | 95.75 |
| 7 | 0 | 1 | 93.95 | 94.46 |
| 8 | 0 | 0 | 98.01 | 95.22 |
| 9 | 0 | −1 | 94.85 | 93.55 |
| 10 | 1 | 1 | 98.01 | 94.96 |
| 11 | −1 | 0 | 93.44 | 95.21 |
| 12 | −1 | −1 | 96.69 | 96.36 |
| 13 | −1 | 0 | 97.76 | 96.56 |
| 14 | 0 | 0 | 95.88 | 95.74 |

### 3.2.2. Establishment of the Regression Model and Analysis of Variance

Design-expert statistical analysis software was used to perform multi-distance regression fitting analysis on the test data in the above table, and the quadratic regression models of the coded values of the stability coefficient of the ditching depth and the consistency coefficient of the ditching bottom width were obtained, as shown in Equations (1) and (2).

$$y_1 = 96.03 - 1.35x_1 - 0.24x_2 + 0.95x_1x_2 + 1.02x_1^2 - 0.016x_2^2 \tag{1}$$

$$y_2 = 97.63 - 0.79x_1 + 0.29x_2 + 0.53x_1x_2 - 1.52x_1^2 - 0.02x_2^2 \tag{2}$$

Thus, a response surface regression model was established for the stability of the trenching depth and the consistency coefficient of trenching width of the forward speed, and the angle between the trenching plough and the ground; the variance analysis was conducted for the regression model. The results are shown in Table 5.

As can be seen from Table 5, the significant *p*-value of the retrospective model of the ditching depth stability and ditching bottom width consistency is less than 0.05, indicating that the regression model is significant. The *p*-values of the misfitting terms were all greater than 0.05, indicating that there was no misfitting factor, indicating that the regression equation had a high fitting degree. The $R^2$ coefficients of the model determination were 0.8736 and 0.9751, respectively, indicating that the model could describe the experimental results well.

**Table 5.** Variance analysis of the regression model.

| Evaluation Indices | Source of Variance | Quadratic Sum | Degrees of Freedom | Mean Square | $p$ | Significance |
|---|---|---|---|---|---|---|
| Stability coefficient of the ditching depth $y_1$ (%) | model | 8.42 | 5 | 4.21 | 0.0172 | * |
| | $X_1$ | 0.21 | 1 | 0.41 | 0.6121 | |
| | $X_2$ | 0.32 | 1 | 0.52 | 0.6014 | |
| | $X_1X_2$ | 4.19 | 1 | 6.67 | 0.0331 | * |
| | $X_1^2$ | 0.44 | 1 | 0.72 | 0.3927 | |
| | $X_2^2$ | 3.26 | 1 | 5.68 | 0.0124 | * |
| | Residual | 3.71 | 5 | - | - | |
| | Loss of quasi item | 3.02 | 3 | 5.02 | 0.0702 | |
| | Pure error | 0.69 | 2 | - | - | |
| | summation | 12.13 | 10 | - | - | |
| Consistency coefficient of the ditching bottom width $y_2$% | model | 16.48 | 5 | 8.52 | 0.0039 | ** |
| | $X_1$ | 4.55 | 1 | 16.43 | 0.0074 | ** |
| | $X_2$ | 0.26 | 1 | 0.74 | 0.5703 | |
| | $X_1X_2$ | 1.44 | 1 | 4.61 | 0.2109 | |
| | $X_1^2$ | 1.27 | 1 | 5.02 | 0.0922 | |
| | $X_2^2$ | 8.96 | 1 | 30.63 | 0.0027 | ** |
| | Residual | 2.67 | 5 | - | - | |
| | Loss of quasi item | 0.91 | 3 | 0.86 | 0.6134 | |
| | Pure error | 1.76 | 2 | - | - | |
| | summation | 19.15 | 10 | - | - | |

Note: * Means significant influence, $p < 0.05$; ** means very significant, $p > 0.01$.

### 3.2.3. Analysis of the Effects of Interaction

According to the regression equation analysis, using the response surface, we investigated the influence of various parameters on the evaluation index, studying speed v, namely, the furrowing plough blade with ground plane angle θ. Two factor interactions are split groove depth stability and the influence of the groove width uniformity; the corresponding surface interaction diagram should be drawn as shown in Figure 9a,b, as shown in the analysis of interaction effects, in order to obtain the optimal parameter combination.

Figure 9a shows the speed and angle of furrowing. The mutual influence of the split groove depth stability coefficient can be seen from the figure; the trenching depth with the increase in the furrowing angle stability declines slowly after the first increases the change trends, and the current speed is at a low level, furrowing angle split groove depth stability coefficient is significant. In addition, the curve changes sharply, indicating that the forward speed varies within the range of 0.9–1.1 m·s$^{-1}$. Appropriately reducing the opening angle can significantly improve the stability of the opening depth. As can be seen from Figure 9a, when the opening angle is constant, the stability of the trenching depth decreases gradually with the increase in the forward speed.

Figure 9b is the speed and angle of the furrowing. The mutual of groove width uniformity coefficient of interaction can be seen in the figure. The current into the speed is constant, as is the groove width uniformity coefficient, which increases the furrowing angle after the first increase with a slowly decreasing trend. When the open groove angle must have a groove width uniformity coefficient, it decreases with the increase in speed; when the trenching angle is at a low level, the forward speed has a significant impact on the consistency of the trench bottom width, which is shown in the figure as the steep curve changes, indicating that when the trenching Angle is suitable for 25–35°, appropriately reducing the forward speed can significantly improve the consistency of the trench bottom width.

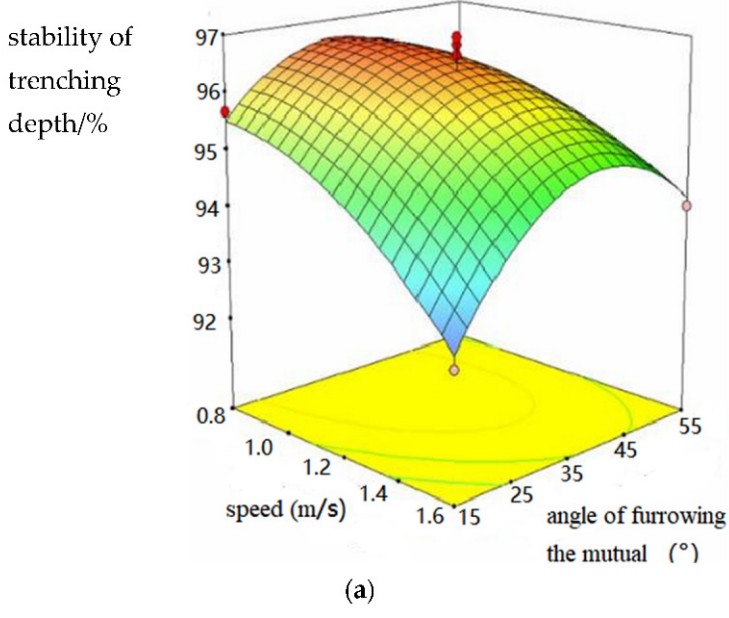

(**a**)

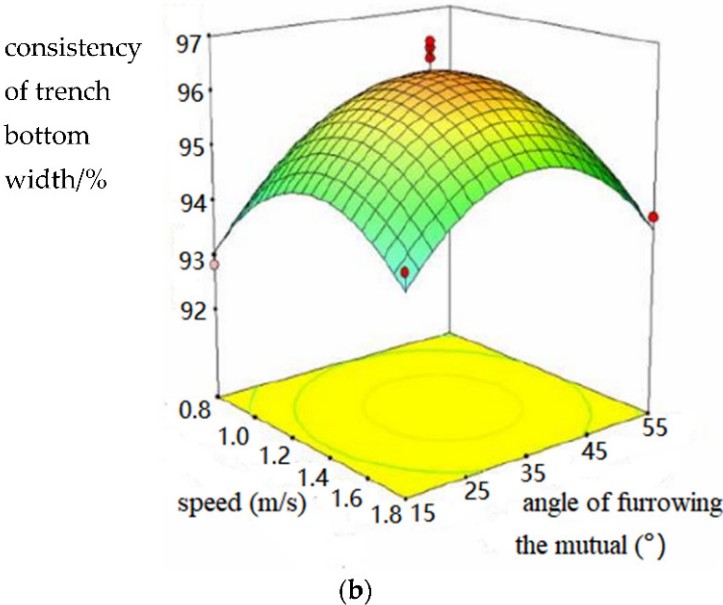

(**b**)

**Figure 9.** Response surface diagram of the interaction. (**a**) Influence of the interaction of trenching depths on the stability. (**b**) Effect of the interaction of trench bottom width on consistency.

### 3.2.4. Optimization of the Parametric Model

According to the agronomic requirements of the ditching operation of Codonopsis pilosula and the actual situation of the entire machine operation, it is required to ensure the stability of the ditching depth and the consistency of the ditch bottom width at the same time to achieve the best results. Because the influence of various factors on its target value was not consistent, a multi-objective optimization solution was needed. Taking the stability of the ditching depth and the consistency of the ditching bottom width as the objective function, the two experimental factors of the forward speed and ditching angle of the ditching device were optimized. The mathematical model is as shown in Equation (1) $\{x_1 \in [0.9, 1.5]\}$ and Equation (2) $\{x_2 \in [15, 45]\}$ in Section 3.2.2.

The influence of the two factors on the stability of the ditching depth and consistency of ditch bottom width was comprehensively considered and optimized and solved

by MATLAB. Finally, the optimal operation parameter combination was obtained: the forward speed was 0.9 m/s, the angle of ditching was 35° and the stability coefficient of the ditching depth and consistency coefficient of ditch bottom width were 97.57% and 98.03%, respectively.

## 4. Discussion

The harvest object of Codonopsis pilosula is the stem. Codonopsis codonopsis is a deep-rooted plant, suitable for viscous soil. Water-logged soil is not beneficial for it. Codonopsis pilosula is mainly implemented with oblique transplanting. To plant it, an oblique ditch must be dug, and the seedlings must be inserted evenly on the sloping surface. Then, cover the seedlings with soil and level it. The main planting area for codonopsis pilotica is the Weiyuan county in the Gansu province, northwest China. The place is called "the home of Codonopsis pilotica in China". At present, there is no sufficient research on the mechanized ditching technology of Codonopsis pilosula, and there is no research on the ditch planting of Codonopsis pilosula in related fields.

In this paper, the key factors affecting the quality of the ditching operation (the stability coefficient of the ditching depth and the consistency coefficient of the trench bottom's width) were analyzed by statistical methods combining EDEM simulation analysis and experiment, single factor test analysis and orthogonal test analysis. The discrete element calibration of the soil particles was completed with EDEM software. The regression model between the test indexes and test factors was established by Design Expert analysis software. we analyzed each factor's influence on the test indices and optimized each experimental factor comprehensively. Finally, we obtained the optimal parameter combination, which provides a technical basis for the mechanization of Codonopsis pilosula's ditching operation.

## 5. Conclusions

(1) An EDEM discrete element simulation model was established in this paper. The simulation results show that the traction resistance of the plough body was affected by the operating speed. The traction resistance increased linearly in low-speed stage and exponentially in the high-speed stage along with an increase in the operating speed. As the simulation speed increased from 0.9 m/s to 1.5 m/s, the traction resistance increased from 1751. 31 N to 2197. 31 N, which was 1.25 times than before. The increase trend gradually increased. It showed that speed had a significant influence on the traction resistance. With the increase in working speed, the traction resistance of the plough body and the corresponding power consumption increased significantly.

(2) Using the stability coefficient of the ditching depth and the consistency coefficient of the trench bottom width as test indices, the speed and the angle of ditching were considered as impact factor. We used a Box–Behnken orthogonal experimental design method to establish a regression model between the test indices and factors, then analyzing the influence of each test index. Combining agronomic requirements and MATLAB's optimal test factors, an optimized operational parameter combination was obtained. The forward speed was 0.9 m/s, the angle of ditching was 35°, and the stability coefficient of ditching depth and consistency coefficient of trench bottom's width were 97.57% and 98.03%, respectively.

(3) The field experiment showed that the evaluation indices of the performance of the adjustable ditching device's performance were in line with the national standards and codonopsis pilosula's planting agronomic requirements. The stability coefficients of the ditch depth and the compartment surface reached 94.67% and 97.53%, respectively, better than the industrial standards.

(4) The paper showed that the adjustable small ditching plough has the characteristics of flexible performance, small power, energy saving, simple operation and convenient operation. The product's application and promotion are relatively speedy.

**Author Contributions:** Conceptualization, D.L. and Y.G.; methodology, X.C. and X.Z. (Xiao Zhang); software, Q.Y.; validation, D.L., X.C. and Y.W.; formal analysis, Y.G.; investigation, D.L.; data curation, Q.Y.; writing—original draft preparation, D.L.; writing—review and editing, D.L.; supervision, X.Z. (Xuejun Zhang) and Y.G. All authors have read and agreed to the published version of the manuscript.

**Funding:** This research was supported by the Central Public-interest Scientific Institution Basal Research Fund, grant number S202101-02; the China Agriculture Research System, grant number CARS-25.

**Institutional Review Board Statement:** Not applicable.

**Data Availability Statement:** The data presented in this study are available on-demand from the first author at (liudejiang@caas.cn).

**Conflicts of Interest:** The authors declare that they have no competing interest.

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
