# Peer review of "EDEM Simulation Study on the Performance of a Mechanized Ditching Device for Codonopsis Planting"

_agriculture, doi:10.3390/agriculture12081238_

Round 1

Reviewer 1 Report

In order to be published, the followings are required: 1) answers to the problems noted on the manuscript; 2) the substantial improvement of the scientific language; 3) the substantial improvement of the English language. See the manuscript with comments in the attached file.  

Author Response

Reply letter

Dear exper teacher,

Thank you for your comments on my paper. I have revised it as required. Please see the attachment PDF.  The PDF file has answered the teacher's question point -by-point. If there is anything wrong, please criticize and correct it. Thank you very much again.

Best wishes

Reviewer 2 Report

General:

Congratulations for interdisciplinary and actual topic, instruments, and results! The conclusions could be interesting for future research and practical activities!

However, there are some issues that must be addressed to improve the overall communication of author`s work:

·       The title, methodology and results could be seriously improved, it seems to be confused and not at all synchronized.

·       Please, use the SI units and consequence in concepts.

·       References must be presented in the same mode all over the paper (or numerical, or author(s) and year).

·       Please, use the same SI units

·       Regarding orthography, starting at title („…of of …”), must be more carefully expressed.

Title is „Experimental studies …”, but in fact is simulation!!! (usually experiment means in field, the paper don't describe in field own experiments)

Abstract must include small ideas about objective of the paper correspondent to method, results, and discussions. 

Introduction

Must be consequent and correlated to method and results.

Introduction must be an actual analysis of the problem. Can be improved by more actually citation.

Please, use SI units and the same type of citation.

Material and Methods

Please, keep in mind that the readers should find here (only and exclusively), detailed description of all your procedures and method (author, process conditions, etc.) used. Anybody must be able to repeat your methods and obtain the same results.

Please, use SI units and the same type of citation. Also, it is recommended to introduce figures in normal order 1 …. n, according with the text description. In text is missing reference to figure 1.

Attention to fig. 2a), text intercalated! (Row 135-136 in old version). Also, correct, please, positions for 2c) and 2d) and description for 2d).

Figure 3 attention to positions of the descriptors! Also, spaces between rows.

In Table 1 and Table 2, please, attention at orthography, Caps at indicators.

Descriptor for fig. 5 and fig. 6 can be improved.

Cap. 2.4.3, please, use SI units.

Results

First part (former Rows 259-263) must be part at methods.

There is a conflict between traction resistance in fig.8 (between 20-27 Nx103) and text (1751.31 N to 2197. 31 N). Also, attention to descriptors in fig 8.

Cap 3.2.1. is part of methodology, including table 3 and 4.

Discussions

In this part must systematically compare own results with others from references and give few interpretation

Conclusions

See observation from results in al.2 about traction resistance.

Author Response

Reply letter

Dear exper teacher:

Thank you for your comments on my paper. I have revised it as required. The blue font is the reply part I modified. If there is anything wrong, please criticize and correct it. Thank you very much again.

Best wishes

Reviewer 3 Report

Dear Authors

All notes and comments are included in the manuscript. The work requires extensive editorial revision. 

Author Response

(The authors gave the same response as above.)

Round 2

Reviewer 2 Report

Thank you for answer and corrections!

Please see with more attention the observations regarding results, especially figure 8 a) and b), where is different description of conditions, and remind kN=1000 N!

Also Title ”... of of ...”, I think can be just one „of”

Thank You!

Reviewer 3 Report

Dear Authors

Thank you for correcting the manuscript and taking into account my comments and questions 
